# The Potential of Metabolomics in the Diagnosis of Thyroid Cancer

**DOI:** 10.3390/ijms21155272

**Published:** 2020-07-24

**Authors:** Margarida Coelho, Luis Raposo, Brian J. Goodfellow, Luigi Atzori, John Jones, Bruno Manadas

**Affiliations:** 1CNC-Center for Neuroscience and Cell Biology, University of Coimbra, 3004-504 Coimbra, Portugal; john.griffith.jones@gmail.com (J.J.); bmanadas@cnc.uc.pt (B.M.); 2Chemistry Department, Faculty of Sciences and Technology, University of Coimbra, 3004-535 Coimbra, Portugal; 3Thyroid Study Group, Portuguese Society of Endocrinology, Diabetes and Metabolism, 1600-892 Lisbon, Portugal; luisraposoendo@gmail.com; 4EPIUnit-Institute of Public Health, University of Porto, 4050-600 Porto, Portugal; 5Department of Chemistry and CICECO, University of Aveiro, 3810-193 Aveiro, Portugal; brian.goodfellow@ua.pt; 6Department of Biomedical Sciences, University of Cagliari, 09042 Cagliari, Italy; latzori@unica.it

**Keywords:** metabolomics, thyroid cancer, biomarker, metabolite, diagnosis

## Abstract

Thyroid cancer is the most common endocrine system malignancy. However, there is still a lack of reliable and specific markers for the detection and staging of this disease. Fine needle aspiration biopsy is the current gold standard for diagnosis of thyroid cancer, but drawbacks to this technique include indeterminate results or an inability to discriminate different carcinomas, thereby requiring additional surgical procedures to obtain a final diagnosis. It is, therefore, necessary to seek more reliable markers to complement and improve current methods. “Omics” approaches have gained much attention in the last decade in the field of biomarker discovery for diagnostic and prognostic characterisation of various pathophysiological conditions. Metabolomics, in particular, has the potential to identify molecular markers of thyroid cancer and identify novel metabolic profiles of the disease, which can, in turn, help in the classification of pathological conditions and lead to a more personalised therapy, assisting in the diagnosis and in the prediction of cancer behaviour. This review considers the current results in thyroid cancer biomarker research with a focus on metabolomics.

## 1. Introduction

Thyroid carcinoma is the most common endocrine malignancy with papillary thyroid carcinoma accounting for 85–90% of all thyroid tumours [1,2,3]. The development of thyroid nodules is a common clinical problem, and its incidence varies geographically worldwide, possibly reflecting environmental influences, the age and sex distribution of the populations studied as well as genetic factors of these populations [4]. The occurrence of these nodules increases with age, is more frequent in females, in iodine-deprived regions and individuals with a history of radiation exposure [5,6]. The estimated annual incidence of thyroid carcinoma worldwide in 2018 was 6.7 cases per 100,000 individuals (https://gco.iarc.fr/today, accessed on 19 June 2019). Furthermore, thyroid cancer affects 567,000 individuals with relatively low annual mortality rates of about 41,000 deaths in 2018 [7].

Thyroid carcinoma develops from two different cell types of the thyroid gland: the follicular and parafollicular cells. Follicular cells are epithelial cells responsible for iodine uptake and thyroid hormone synthesis, while parafollicular cells produce and secrete the hormone calcitonin. Of these two, follicular cells are responsible for most of the thyroid-derived carcinomas [8]. Thyroid carcinoma can be subdivided into papillary, follicular, anaplastic and medullary.

Thyroid nodules can be clinically detected by palpation or thyroid ultrasound. The most widely used clinical method for the diagnosis of thyroid nodules is cytological analysis through fine-needle aspiration biopsy (FNAB), but this procedure often produces inconclusive results. The Bethesda classification divides the results of this cytological examination into six classes: non-diagnostic or unsatisfactory (I), benign (II), atypical or follicular lesions of undetermined significance (AUS/FLUS) (III), suspicion of follicular neoplasia (IV), suspicion of malignancy (V) and malignancy (VI). When the cytological analysis is non-diagnostic (I) or in the case of AUS/FLUS (III), FNAB should be repeated. Scheduling a new FNAB may not always help to clarify the diagnosis, since it could continue to be non-diagnostic or indeterminate, requiring a clinical decision of active surveillance or surgery based on little information. However, when the result is the suspicion of follicular neoplasia or suspicion of malignancy, surgical removal of tissue is recommended for a more complete histological diagnosis [9]. Furthermore, less than 30% of lesions classified as atypical or follicular lesions of undetermined significance (AUS/FLUS) (III), suspicion of follicular neoplasia (IV) and then diagnosed by histology following surgery are subsequently classified as malignant [10], meaning that several surgeries could have been avoided, had the initial diagnosis been conclusive in ruling out malignancy. Moreover, even though histopathological evaluation remains the gold standard in distinguishing different types of thyroid cancer, it may also not be conclusive in some cases [11].

The discovery of standalone malignancy biomarkers is therefore of particular importance for patients with indeterminate cytologies, mainly categories AUS/FLUS (III), suspicion of follicular neoplasia (IV) and suspicion of malignancy (V). Thus, new biomarkers could be useful in the selection of cases for surgery. The need to improve the diagnostic power of FNAB has been a longstanding concern, reflected in an editorial by Ernest L. Mazzaferri in 1992 which reported the uncertainty in a diagnosis performed solely by this biopsy, and emphasised the importance of combining its data with good clinical history [12].

In the last decade, the different “omics” played an essential role in improving the knowledge of complex biological systems and identifying potential new biomarkers of pathological progressions, or pharmacologic responses to a therapeutic intervention. The “omics” field encompasses a large variety of approaches, from genomics (genetic mapping and DNA sequencing of genomes) and transcriptomics (gene expression patterns) to proteomics (protein expression and interactions) and metabolomics (metabolites and related biochemical reactions), which aim to identify biomarkers.

The mitogen-activated protein kinase (MAPK) and phosphoinositide 3-kinase (PI3K) signalling cascades are the main activator pathways implicated in thyroid cancer. Papillary carcinoma, in particular, is mostly related to mutations that activate the MAPK signalling pathway, including *RET/PTC* rearrangements and point mutations of the *BRAF* and *RAS* genes. Genetic alterations found in follicular carcinomas, the second most frequent type of thyroid cancer, include *RAS* mutations and *PAX8-PPARγ* rearrangement [13]. In terms of genetic biomarkers, the *BRAFV600E* mutation has been found to be very specific for papillary thyroid carcinoma; however, its absence cannot reliably rule it out [11,14]. Indeed, these common genetic mutations are considered poor prognostic markers, although they can still be useful at an individual level [15]. The differential expression of micro RNA molecules (miRNAs) in distinct histopathological tumour types and at various stages of tumour differentiation and progression has also been reported to have diagnostic and prognosis potential. MicroRNAs are small non-coding RNAs that can regulate gene expression at the post-transcriptional level and are involved in a wide range of biological processes, including cancer. In thyroid cancer, in particular, the levels of miRNA-146b, -221 and -222 were found to be changed [16,17,18]. Either alone or as a panel, miRNAs could be useful diagnostic and prognostic markers, and may even be considered as therapeutic targets.

While genetic expression corresponds to protein synthesis, it does not necessarily correlate with the concentration of the expressed protein since this depends on its degradation rate as well as its rate of synthesis. Hence. It is important to study not only genomics but also the proteomics of the system. Given the recent mass spectrometry (MS) advances, especially for proteomics applications, this approach has become the gold standard for protein identification. In particular, MS imaging of protein expression has become an interesting technique for future clinical applications since it can be applied directly to intact tissue samples, thereby revealing the spatial distribution of individual proteins. The number of proteomics studies in thyroid cancer currently exceeds those based on metabolomics, and these have been extensively reviewed [19,20]. In particular, protein S100-A6 was identified as a factor that discriminated between follicular and papillary thyroid carcinoma, with overexpression in the latter [21,22,23]. However, its sensitivity and specificity for distinguishing between benign and malignant thyroid cancer were relatively poor (85% and 69%, respectively). Another protein that was found to be upregulated in papillary thyroid carcinoma in comparison to follicular thyroid cancer was 14-3-3 σ [24,25,26]. This protein usually acts as a negative regulator of the cell cycle, but it is thought to have an enabling role in papillary carcinoma. Proteomics per se can provide not only diagnostic and prognostic biomarkers but also reveal potential therapeutic targets. For example, protein HSP90 was found to be overexpressed in thyroid cancer [27,28]. This protein is responsible for the folding of many proteins directly associated with malignant progression, so its inhibition could result in a combinatorial attack on numerous oncogenic pathways. Inhibition of HSP90 can therefore not only attenuate cell proliferation but also increase the efficacy of radioiodine therapy in thyroid cancer patients [29,30,31,32], making this protein a good chemotherapeutic target.

Recently, a proteomics profiling of tissues for the four types of thyroid cancer and benign follicular adenoma revealed that several proteins associated with metabolism, including mitochondria-related functions, lipid and nucleic acids metabolism, could discriminate between these different thyroid lesions [25]. Besides being capable of discriminating between cancer types, the integration of these results with metabolomics and transcriptomics data has enabled a more complete understanding of the pathogenesis of these cancers. This is a good example of how the integration of multiple omics approaches can provide a more holistic understanding of thyroid cancer, as well as confirm the mechanistic insights obtained from each omic strategy.

Metabolomics carry a great promise for discovering potential markers of various pathological conditions and for better understanding biochemical pathways associated with disease development [33,34,35]. The metabolome comprises the entire set of small molecules that are generated by metabolic activity within cells, tissues and organs. Its composition may change in response to enzyme levels and activities, cellular regulation, signalling pathway activation, and genetic variations. The metabolome can reflect changes in the transcriptome (mRNA) and proteome (proteins), as a consequence of environmental factors, such as drugs, nutrients and pollutants, in addition to phenotypic alterations in pathophysiological states. In particular, metabolomics provides a phenotypic snapshot of a cell, tissue or organism, reflecting more closely the clinical reality, thus improving the understanding of physiopathological mechanisms. As metabolites are the end-products of biochemical reactions in the body, they are the closest molecules to phenotype. On this basis, metabolomics could represent a very useful tool for the identification of metabolic pathways specific to thyroid cancer, alongside other omics techniques. Since tumours significantly alter primary metabolic pathways, metabolomics is rapidly becoming an important approach to identify cancer biomarkers too. Alterations of the metabolome can also be mirrored in different biofluids such as blood [36], urine [37] and amniotic fluid [38]. Thus, analysis of a biofluid can also be alternative/integrative data to FNAB. In the last few years, the capability for determining metabolic profiles through the use of nuclear magnetic resonance (NMR) spectroscopy or mass spectrometry has grown significantly, resulting in a consequent increase in metabolomic studies of thyroid cancer. However, more studies are needed to identify possible biomarkers and better understand the mechanisms involved in the metabolic alteration in thyroid cancer. In this review, we discuss how metabolomics has been used to study thyroid cancer by focusing on the original papers in the matter and suggest future perspectives.

## 2. Metabolomics in Thyroid Cancer

Compared to proteomics and transcriptomics, thyroid cancer metabolomic studies have featured in relatively few papers over the last ten years. However, recent technical improvements in both hardware (mass spectrometers with higher mass accuracy, SWATH data-independent MS acquisition and ion-mobility MS) and software (improvement of metabolite identification databases, as well as metabolite biological integration and NMR automatic identification and quantification) have allowed metabolomics to emerge as a standalone method for profiling of thyroid cancer samples [39,40,41]. Although this review is focused on NMR and MS metabolomics, it is worth mentioning that other techniques could also be applied. Raman spectroscopy also has an interesting diagnostic potential: by analysing the vibrational modes of chemical bonds, it can identify non-specific molecules, such as proteins, lipids or nucleic acids, that may just be enough to distinguish between malignant and benign samples [42,43].

### 2.1. The Early Years—NMR Spectroscopy

One of the first metabolomic studies that attempted to address the lack of diagnostic power in thyroid cancer was in 1994 and consisted of a ^1^H NMR study of 19 malignant and 24 benign patient tissue samples (Table 1). The authors were able to identify triglycerides and lysine as potential discriminatory metabolites, but the method’s specificity was only 52% [44]. Two years later, the same authors applied two-dimensional NMR spectroscopy, which improved the resolution of metabolite signals, allowing a higher number of metabolites to be monitored. However, this only led to a moderate improvement in the method specificity [45].

By the beginning of the 21st century, NMR spectroscopy had emerged as the main technique for performing metabolomic analysis. The first proof-of-concept studies in thyroid cancer used either magnetic resonance spectroscopy imaging (MRSI) [46,47,48] or ^1^H NMR spectroscopy on excised tissue samples or deproteinised tissue extracts [49,50]. One interesting feature of the study of King et al. is that it was one of the first studies to identify choline as a metabolite whose levels were changed in thyroid cancer [46]. This was confirmed in subsequent studies and choline has since often been proposed as a thyroid cancer biomarker. However, it should be emphasised that, while MRSI is non-invasive, the standard 1.5T systems in current clinical use are limited to detecting a handful of highly abundant metabolites, such as choline, within relatively large voxel volumes (≥1 mL) [46,47,48].

The first high-resolution magic angle spinning (HR-MAS) NMR metabolomics study [51] and the first MS study [52] that we found in our literature search were both published in 2011. HR-MAS allows spectra to be obtained from intact biopsy samples of 10–40 μL volumes with signal resolution approaching that of high-resolution NMR spectra of tissue extracts. The study of Jordanet al., although using a limited number of samples, had the benefit of being able to compare results of tissues with those of aspirates. The study of Yao et al. analysed the serum of 30 papillary thyroid carcinomas (malignant), 80 nodular goitres (benign) and 30 healthy controls and found that malignant and benign samples were correlated with changes in lipid metabolism, with 3-hydroxybutyric acid, an intermediate product of fatty acid metabolism, particularly important. One year later, the group of Prof. Caldarelli published two similar papers [53,54] using HR-MAS NMR on tissue samples. These revealed increased phenylalanine, taurine and lactate levels, and a decrease in choline and choline derivatives and *myo-* and *scyllo-*inositol levels in malignant tissues compared to benign. However, when these data were modelled using orthogonalised partial least-squares discriminant analysis (OPLS-DA) their diagnostic power was found to be limited, as indicated by the area under the curve (AUC) of the receiver operating characteristic (ROC) of 0.77 [55].

In another study, ^1^H HR-MAS NMR of tissue, in conjunction with ^1^H NMR from plasma samples, was used to classify papillary thyroid microcarcinomas, a subtype of papillary carcinoma. By using nine significantly changed metabolites from plasma (glucose, mannose, pyruvate, 3-hydroxybutyrate, valine, tyrosine, proline, lysine and leucine), they were able to achieve good sensitivity and specificity with an AUC of 0.992 [56]. This technique was more recently used in FNABs of thyroid tissues collected post-surgically and found statistically relevant metabolites in indeterminate lesions (*myo-* and *scyllo-*inositol, serine, citrate, leucine, alanine, phenylalanine and tyrosine) [57]. While HR-MAS NMR can provide ^1^H NMR spectra of semisolids with a comparable spectral resolution to liquid-state NMR, it requires high spinning rates of several kHz. This may not only disrupt the tissue structure but can also result in the leakage of potentially infectious material. Furthermore, HR-MAS probes are costly, while incomplete suppression of the water signal can also interfere with the quantification of some metabolites. Therefore, ^1^H-NMR spectroscopy of tissue extracts has continued to be widely used [56,58,59,60,61,62,63]. In the study of Deja et al., four metabolites were considered as selective biomarkers of thyroid cancer, namely creatine, *myo-* and *scyllo-*inositol and uracil, but the thyroid cancer group was comprised of only 12 patients [58]. The study by Metere at al., although with only 14 patients, observed differences in cancer and healthy tissue in lactate, phenylalanine, citrate, myo-inositol and threonine [63]. Tian et al. were able to distinguish malignant thyroid lesions from benign with a ROC of 0.88 [59]. On the other hand, Ryoo et al. from the aspirates alone could distinguish seven metabolites (lactate, choline, O-phosphocholine, glycine, citrate, glutamate and glutamine) with ROCs ranging 0.64–0.85 [60]. Seo et al. attempted to predict lymph node metastasis in papillary carcinoma patients, but they were not able to discriminate the presence of metastasis [61]. In the study of Li et al., 15 metabolites were found to be differentiated using two OPLS-DA models [62].

Although NMR spectroscopy has been a valuable technique for several metabolomic studies so far, it has had a strong competition by MS in the last few years. One of the reasons is its lower sensitivity in comparison to MS. However, NMR spectroscopy presents advantages in relation to MS, by being highly reproducible and capable of performing absolute quantification of the metabolite’s concentrations. Furthermore, it can detect compounds that are less easily detected by MS, such as sugars, organic acids, alcohols and other highly polar compounds, and it is well suited for studying intact tissues, organs and other solid or semi-solid samples through solid-state NMR and HR-MAS NMR. However, metabolite identification is not straightforward given the complexity of the ^1^H-NMR spectra but can be more easily overcome by databases such as the Human Metabolome Database (HMDB) [64], or the use of (semi)automatic identification and quantitation tools such as BAYESIL [65] or Chenomx NMR Suite from Chenomx Inc.. This complexity comes mainly from peak overlap, which could be ameliorated by the use of stronger magnets, increasing spectral dispersion. Presently, commercial NMR spectrometers can achieve magnetic fields of 28.2 Tesla, the equivalent to a ^1^H Larmor frequency of 1.2 GHz, but unfortunately, the cost of such equipment is by now detrimental to their use, with the 600-MHz NMR spectrometers being the best cost-sensitivity/resolution compromise. The more frequent application of selective excitation techniques on specific spectral regions and of multidimensional NMR experiments such as total correlation spectroscopy (TOCSY) and J-resolved spectroscopy (J-Res) [56,59,66] could also help in resolving overlapping peaks. Another exciting development in NMR metabolomics is in probes design, with microprobes for MAS enabling an enhancement of sensitivity while reducing the sample size to a few microliters, and cryoprobes significantly increasing signal sensitivity.

### 2.2. The Rise of Mass Spectrometry

Even though the sensitivity of NMR spectroscopy has significantly improved over the last few years with a metabolite quantification threshold of ≥1 µM, it remains far less than that of MS [67]. With the improvements in instrumentation, experimental methods, software and spectral databases, the use of mass spectrometry in the field of metabolomics has grown considerably in recent years, including its application to metabolomics studies of thyroid cancer (Figure 1). Liquid chromatography coupled to mass spectrometry (LC-MS) was first used to study thyroid cancer in serum samples from 30 papillary thyroid carcinoma, 80 benign thyroid nodules and 30 healthy controls [52]. 3-hydroxybutyric acid, an intermediate product of fatty acid metabolism, was found to be higher in the papillary thyroid carcinoma group compared to either benign or healthy groups. In 2017, Zhou et al. applied a data-independent acquisition (DIA) workflow for metabolomics [68]. Unlike the traditional data-dependent acquisition (DDA) strategies, this acquisition mode has higher metabolite coverage by using mass range windows to obtain the fragmentation spectra. It is expected that this innovative way of LC-MS metabolite profiling will be translated into metabolomic studies [69]. However, alternative approaches can be used in thyroid cancer metabolomics; for example, an amino acid analyser was used on the plasma of thyroid cancer patients and found significantly higher levels of methionine, leucine, tyrosine and lysine [70]. In addition to the analysis of water-soluble metabolites, several lipid species have also been identified as putative biomarkers for resolving malignant and benign thyroid lesions. Ishikawa et al. combined imaging mass spectrometry with a matrix-assisted laser desorption/ionisation tandem time-of-flight (MALDI-TOF/TOF) instrument to identify and describe the distribution of individual biomolecules in a tissue section [71]. With this approach, they revealed that phosphatidylcholine (34:1) and (34:2) and sphingomyelin (34:1) were present in significantly higher amounts in papillary thyroid carcinoma when compared to normal tissue from the same patients. A similar approach was applied to tissue and serum samples collected from subjects with malignant or benign lesions (tissue), as well as healthy subjects with no thyroid lesions (serum). In this case, it was found that a biomarker panel consisting of phosphatidic acid (36:3) and sphingomyelin (34:1) could distinguish malignant cancer from benign, with an AUC value of 0.961, a sensitivity of 87.8% and a specificity of 92.3% [72]. Zhang et al. observed increased relative abundances of ceramides and specific glycerophosphoinositols using 2D desorption electrospray ionisation mass spectrometry to image thyroid cancer in lymph node tissues [73]. Meanwhile, Huanget al. showed a higher expression of phenylalanine, leucine and tyrosine in the tumour region with a gradual level decrease from tumour to the stromal and normal tissues and the inverse profile of creatinine [74]. Another study was able to profile lipids directly in formalin-fixed tissue sections by MALDI-Q-Ion Mobility-TOF-MS, demonstrating that this technique could be complementary to the present histological methods [75]. These studies demonstrate the potential of spatially resolved metabolomics to provide meaningful and clinically relevant information from biopsy samples that are by nature highly heterogeneous.

The year 2015 saw a peak in the number of metabolomic publications, with gas chromatography-mass spectrometry (GC-MS) being widely reported (Figure 1). This technique was first used in combination with a ^1^H NMR metabolomics study to measure fatty acid abundances [59]. They showed higher levels of (C14:0), (C16:0) and (C18:3n3) fatty acids and lower levels of (C20:3n6) fatty acids in malignant compared to benign tissues. Since then, other GC-MS metabolomics studies have been published that identified metabolites in carbohydrate metabolism, including glucose, fructose, galactose, mannose, 2-keto-D-gluconic acid and rhamnose that were decreased in papillary thyroid carcinoma, which is consistent with an upregulation of the glycolysis and pentose phosphate pathways [76]. These results are consistent with cancer tissues requiring higher rates of cytosolic ATP production and increased amounts of NADPH and precursors for biosynthesis of nucleotides and other cell components. Another study combined the metabolic profiles obtained by GC-MS and ultra-performance liquid chromatography-mass spectrometry (UPLC-MS), resulting in a total of 195 detected metabolites. From these metabolites, they concluded that purine and pyrimidine metabolism was higher in papillary thyroid carcinoma, as well as taurine and hypotaurine levels. However, another study that used GC-MS and UPLC-MS identified a decrease in galactinol, melibiose and melatonin in papillary thyroid carcinoma with an AUC of 0.96 [77].

In an attempt to discriminate between different types of thyroid cancer, and some of their most common variants, Wojakowska et al. analysed five different types of thyroid malignancies (follicular, papillary classical variant, papillary follicular variant, medullary and anaplastic cancer), as well as benign follicular adenoma and normal thyroid tissue [78]. They found an upregulation of lactic acid and downregulation of several fatty acids and their esters in cancer versus normal tissue, as well as upregulation of *myo-*inositol phosphate, succinic acid and certain fatty acids and their esters in malignant versus benign tissue. Moreover, the classical variant of papillary carcinoma could be distinguished from follicular thyroid lesions by lower levels of gluconic acid and higher amounts of citric acid. In addition, follicular carcinoma could be distinguished from the follicular variant of papillary carcinoma by changes in the levels of decanoic acid ester. It would be important to promote more studies which discriminate between different types of thyroid cancer since cancer classification is essential to assess prognosis and select an adequate treatment. Moreover, follicular adenoma, follicular carcinoma and the follicular variant of papillary carcinoma can be hard to distinguish histologically, so metabolomics can represent an important tool to assist in their differentiation. Regarding more specific studies, the serum of 37 patients with distant metastasis was compared with the serum of 40 patients from an ablation group, where it was found that serum asparagine, gamma-amino butyric acid (GABA), aminooxyacetic acid and 4-deoxypyridoxine increased in the distant metastasis group while pyroglutamic acid was decreased [79]. A GC-MS metabolomic study was also performed on a model system of thyrospheres, containing cancer stem-like cells, from B-CPAP and TPC-1 cell lines derived from papillary thyroid cancer of the BRAF-like expression profile class, which showed significant differences in Krebs cycle intermediates, amino acids, cholesterol and fatty acids content when compared to non-cancer stem-like cells [80]. Besides in vivo measurements, it may be interesting to characterise individual cell types found within the tumour, given the heterogeneity of cancer cells, to therapeutically target those that are contributing the most to the cancer phenotype. The papillary thyroid cancer-derived cells also showed altered redox homeostasis as well as increased levels of intracellular oxidant species, a common hallmark of cancer, since ROS homeostasis needs to be tightly regulated, otherwise it can promote an altered metabolism. The most perturbed metabolic phenotype was found in B-CPAP cells, which are characterised by the most aggressive genetic background [81], demonstrating the connection between genetic background and cancer metabolism and consequently phenotype. Once again, we observe the importance of combining information from genetics to metabolism for a better understanding of this disease.

The field of mass spectrometry-based metabolomics has been facing a significant evolution with more sensible, higher dynamic range, higher data acquisition speeds and different acquisition modes equipment. Nonetheless, data acquisition is not the only critical point. Data analysis with better algorithms for peak detection, alignment and analysis; better software tools that integrate these algorithms and further statistical analysis; and better databases with information on each compound, such as possible adducts and multiple retention times (XCMS/Metlin and HMDB), are pushing the field forward at higher speeds. More specifically, identification of metabolites on a large scale with the assistance of software tools (Elucidata El MAVEN [82] and Sciex Accurate Mass Metabolite Spectral Library with MasterView™ software) or using sample preparation kits (IROA^®^ Quantitation Kits) will advance even further mass spectrometry as the go-to methodology for metabolomics. Thyroid cancer profiling, in particular, will definitely benefit from these advances. Moreover, the technical advances for mass spectrometers have allowed even for their use in the clinical setting. Take, for example, an automated and biocompatible handheld mass spectrometer that can quickly and non-destructively assess if at the pointed location cancerous tissue is present, which allows surgeons to accurately define the tumour margins prior to excision [79].

### 2.3. Peripheral Fluids

Most of the publications for thyroid cancer metabolomics to date have focused on the direct analysis of the thyroid gland. However, with the aim of avoiding the invasive biopsy of the thyroid, there have also been studies that looked for associations between thyroid malignancies and plasma [50,56,70,83] or serum metabolites [52,72,79,84,85]. A study of children and adolescents with thyroid cancer identified an increase in the levels of serum leucine, lactate, alanine, lysine, acetate, glycine and choline and lower levels of glucose in papillary thyroid carcinoma samples versus benign by ^1^H NMR spectroscopy, which is consistent with other works in adults [86]. More recently, a plasma GC-MS study has suggested sucrose as a discriminative compound between papillary thyroid cancer and multinodular goitre, which poses an interesting question as to the influence of high sucrose sugar diets in the promotion of tumorigenesis [83]. Another non-invasive approach used capillary electrophoresis to analyse urine [87]. In this case, the authors focused on the profiling of urinary nucleosides, with inosine, N^2^-methyl guanosine, N^2^-N^2^-dimethylguanosine and 1-methylguanosine being higher in thyroid cancer patients when compared to healthy controls. In a study of paired blood and urine samples, it was shown that, while metabolome data of each analyte could differentiate between healthy subjects and those with nodular lesions, analysis of the combined datasets provided better predictive power [88]. Another study, collecting both serum and urine, indicated serum β-hydroxybutyrate, docosahexaenoic acid, 1-methyladenosine, pregnanediol-3-glucuronide, urinary nicotinic acid mononucleotide and xanthosine as a potential biomarker panel for papillary thyroid cancer, using two validation sets [89]. Huang et al. integrated data of serum and plasma metabolites from six independent centres, having had a total of 1540 serum-plasma matched samples and 114 tissues [90]. The study was divided into a discovery phase, composed of one centre and then the validation phase, with the rest of the samples from the other centres. They were able to establish a panel of six biomarkers with an AUC of 98%, namely *myo-*inositol, α-N-phenylacetyl-L-glutamine, proline betaine, L-glutamic acid, lysophosphatidylcholine (18:0) and lysophosphatidylcholine (18:1), to distinguish between healthy samples and papillary thyroid carcinoma. However, they were not able to distinguish cancer samples from benign thyroid nodules. Another study compared the plasma lipidomic profiles of five commonly found cancers: liver, lung, gastric, colorectal and thyroid [91]. Interestingly, they found a distinct profile in thyroid cancer relative to all the other studied cancers, selecting lysophosphatidylinositol (18:0) and (18:1) as specific to thyroid cancer only. Going beyond blood and urine, a study of thyroid carcinoma patients and healthy controls revealed highly predictive differences in intestinal microbiota genera and faecal metabolites [92]. Finally, exhaled breath from 39 papillary thyroid carcinoma, 25 benign and 32 healthy volunteers was analysed by solid-phase microextraction GC-MS with (3-methyl-oxiran-2-yl)-methanol, 1,1,3-trimethyl-3-(2-methyl-2-propenyl) cyclopentane and trans-2-dodecen-1-ol being identified as significantly changed in papillary thyroid carcinoma versus benign [93].

The study of body fluids is important to give a broader overview of the disease depending on the compartmentalisation of such fluids. For example, urine is highly dependent on food and liquid intake, while blood can have a more stable metabolome. Biofluids imply non-invasive or minimally invasive collection when compared to tissues and reflect the overall response of the patient to the disease. They have therefore the potential to be used in the monitoring of therapy and cancer’s evolution. In the case of thyroid cancer diagnosis, it would be important to find a minor invasive method that could complement the FNAB exam and would ultimately provide a faster and more accurate diagnosis of thyroid cancer. Furthermore, omics profiles in these samples can bring us closer to precision medicine, where an individual’s metabolomic fingerprint can assist the physician in therapy customisation.

### 2.4. Most Referred Metabolites

The most cited metabolites in thyroid cancer profiling are identified in Figure 2. The top three are choline, lactate and tyrosine. When comparing different studies, the differences in the abundance of these metabolites between healthy subjects or patients with benign and patients with malignant thyroid lesions do not always match. This might be explained by the use of different sampling methods, techniques and study groups. Nevertheless, there is a broad consensus that lactate is upregulated in thyroid cancer. According to Warburg’s hypothesis, cancer cells present a high glycolytic rate with high rates of glucose conversion to lactate even with plentiful levels of oxygen [94]. Although far less ATP is generated per mole of glucose via glycolysis, compared to oxidative phosphorylation, it is generated at a much faster rate [95]. Moreover, the lactate product may not be a bystander in this process, because it is thought to have a role in angiogenesis, immune escape, cell migration, metastasis and self-metabolism [96]. Thus, it is not surprising that lactate has been identified as a biomarker in thyroid cancer as well as in lung [97], breast [98] and pancreatic cancer [99].

Tyrosine is considered a non-essential amino acid because it can be synthesised from phenylalanine; nonetheless, it has an important role in the production of proteins that are a part of signal transduction processes, acting as a receiver of phosphate groups transferred through tyrosine kinases. In turn, these enzymes have been associated with the regulation of cellular proliferation, survival, differentiation, function and motility, linking them to a cancer phenotype [100]. In particular, thyroglobulin, which contains multiple tyrosine residues, is a protein produced by the follicular cells of the thyroid, and that is, in turn, the precursor of thyroid hormones T3 and T4. The dysregulation of thyroglobulin is associated with multiple thyroid diseases, including cancer. Besides tyrosine’s role in protein biosynthesis, it acts as an intermediate in the biosynthesis of catecholamines, which have also been associated with thyroid cancer [101]. The role of choline and choline derivatives is not clear since some studies have shown that they are downregulated [50,53,54], while others have shown them as upregulated [46,47,59,60,71]. Nonetheless, choline has also been associated with other types of cancer [102]. Activated choline metabolism, characterised by increased phosphocholine and total choline-containing compounds is a common hallmark of different cancers [103] and increased levels of choline and its derived compounds have been associated not only with proliferation but also with malignant transformation [104]. Other metabolites that are mentioned frequently could even be explained by the same mechanisms and cancer driven pathways. This is the case of lactate, citrate, alanine, glutamic acid, glutamine and leucine, since their dysregulation coincides with glutaminolysis in a cancer context. Glutamine is therefore heavily consumed to generate ATP and lactate, as well as being used in the synthesis of other molecules, such as nucleotides and proteins, producing during these mechanisms more glutamic acid, alanine and leucine. Citrate on the other hand, will be likely consumed to provide acetyl-CoA for lipid synthesis.

Cancer cells reprogram their metabolism to meet high energy needs and require increased biosynthesis to be able to grow and divide. The overall dysregulation in lipids and amino acids, as observed in Figure 2, can be associated with a higher metabolic turnover of these species. In the case of lipids, there is a high demand for membrane biosynthesis in cell propagation, which explains why they are decreased in cancer samples, while for amino acids there is possibly a high protein turnover which may explain their unusually high values in malignant tissues.

The metabolites referenced the most in thyroid cancer are also a reflection of the methods used to detect them since these three top mentioned molecules reflect metabolites more easily identified by NMR spectroscopy. Although being mentioned more often, they have not been validated as specific thyroid cancer biomarkers, which shows the importance of using more sensitive techniques, such as mass spectrometry, which could provide other metabolites to be considered as biomarkers. Furthermore, some biomarkers where found decreased in some works, but increased in others. Although significantly different levels of these biomarkers in patients with thyroid cancer versus those with benign lesions could be applied for thyroid cancer diagnosis, it is important to also correlate these changes with tumour growth and proliferation and confirm that differences in biomarker levels correspond to different cancer stages. Moreover, validation of the potential biomarkers found will be crucial. In an exploratory study, cross-validation is enough, but an independent validation, and ultimately the approval of regulatory agencies, is recommended for clinical biomarker establishment. This could be performed through metabolomic data collection in multicentre studies; however, it is extremely important to develop and strictly follow the same standard operating procedures to ensure that data can be combined. The compilation of the literature results in systematic reviews can also aid in confirming the most significant biomarkers for thyroid cancer; for example, Khatami et al. suggested citrate and lactate [105]. It is only by cross-validating information that these biomarkers can be incorporated into clinical decision making. Additionally, two distinct paradigm shifts should be changed in the reporting of metabolomics results: the possibility to submit metabolic fingerprints (such as Raman spectroscopy) instead of well-identified metabolites as biomarkers, and integration of different biomarker types (metabolites, proteins and miRNA) to improve the differential diagnosis of thyroid lesions. In the future, it will also be important to interpret such results in light of the changes in biochemical signalling activity and metabolic pathway fluxes that characterise tumour malignancy. For example, information from online resources such the Kyoto Encyclopedia of Genes and Genomes (KEGG) Pathway Database [106] and Metaboanalyst [107], could be coupled with stable isotope tracer measurements of metabolic fluxes that are upregulated in malignancy, such as glycolytic lactate production [96] and generation of nucleoside precursors via the pentose phosphate pathway [108].

**Table 1 ijms-21-05272-t001:** Summary of metabolomics studies on thyroid cancer in the last 25 years.

Technique	Method	Study Site	Sample	Study Design	Altered Metabolites	Reference
NMR	MRS	Spatially resolved information	Non-invasive method	8 TC vs. 5 CTR	Ch ↑	[46]
	MRS		Non-invasive method	8 MN vs. 17 BN	Ch ↑	[47]
	MRS		Non-invasive method	8 PTC vs. 18 BN	Ch ↑	[48]
	^1^H-NMR	Tissue	Tissue	19 MN vs. 24 BN	TGL, K ↑	[44]
	2D ^1^H-NMR		Tissue	32 MN vs. 61 BN	Cross peaks from CHL/cholesteryl esters and di-/TGL ↑; two unassigned cross peaks ↓	[45]
	HR-MAS-NMR		Tissue	Tissue: 72 TC vs. 28 CTR; Tissue: 38 MN vs. 34 BN; Aspirate ex vivo: 12 TC vs. 12 CTR	Tissue TC vs. CTR: F, Y, S, K, TAU, Q, E, A, I, L and V ↑; Lip ↓. Tissue MN vs. BN: LAC and TAU ↑; Lip, Ch, PC, myo- and scyllo-IST ↓	[53]
	HR-MAS-NMR		Tissue	38 MN vs. 34 BN	F, TAU and LAC ↑; Ch and Ch derivatives, myo- and scyllo-IST ↓	[54]
	HR-MAS-NMR		Tissue	52 MN vs. 46 BN	Y, S, A, L, F↑; myo- and scyllo-IST and CIT ↓	[57]
	^1^H-NMR	Tissue extracts	Tissue extracts	15 TC vs. 19 BN and 27 CTR	CHL ↑; DLC ↓	[49]
	^1^H-NMR		Tissue extracts	45 thyroid lesions vs. 19 CTR from the same participant	M, A, E, G, LAC, Y, F and HPX ↑; ACT ↓	[58]
	^1^H-NMR		Tissue extracts	32 LNM vs. 20 absence of LNM; 19 lateral LNM vs. 33 absence of lateral LNM	No statistically altered metabolites	[61]
	^1^H-NMR		Tissue extracts	16 PTC vs. 16 CTR from the same participant	L, V, G, TAU, LAC, Ch, ETA, GPC and LDL↑; CIT, VLDL ↓	[62]
	^1^H-NMR		Tissue extracts	11 TC vs. 10 CTR from the same participant	LAC, F ↑	[63]
	^1^H-NMR	FNAB	Aspirates	34 PTC vs. 69 BN	LAC, Ch, O-PC, G↑; CIT, E, Q ↓	[60]
	^31^P-NMR	Systemic profiling	Plasma	16 MN vs. 17 hypothyroid in remission and 14 euthyroidism in remission and 23 healthy euthyroid controls	MN vs. hypothyroid in remission: PE + SM and PC ↓	[50]
	^1^H-NMR		Serum	20 PTC vs. 20 BN and 20 CTR	PTC vs. CTR: V, L, I, LACA, A, E, K, G ↑; Lip, Ch and Y ↓	[84]
	^1^H-NMR		Serum	17 PTC vs. 17 BN and 20 CTR	PTC vs. BN: KYN, HIP, NIC, XNT ↑; Q, CIT, O-ALC, GSH, W, Y, HoS, β-A ↓ PTC vs. CTR: myo- and scyllo-IST, W, PPN, LAC, HoC, 3-Me GTA, N, D, Ch ↑; ACM ↓	[85]
	^1^H-NMR		Serum	41 PTC vs. 55 BN and 40 CTR	L, LAC, A, G, K and Ch ↑; GLU ↓	[86]
	^1^H-NMR		Serum and urine	17 PTC vs. 33 BN and 17 CTR	PTC vs. CTR: Serum: CRE ↑; V, A, CRN and Y ↓; Urine: CIT and ACT ↓	[88]
	HR-MAS-NMR	Combination	Tissue and Aspirates	4 PTC, 4 FA, 5 CTR	NA	[51]
	HR-MAS-NMR and ^1^H-NMR		Tissue and plasma	Tissue: 16 PTMC vs. 11 CTR tissues from the same participants; Plasma: 26 PTMC vs. 17 CTR volunteers	Tissue: F, Y, LAC, S, C, K, Q/E, TAU, L, A, I and V ↑; FA ↓. Plasma: same as tissue as well as GLU, MAN, PYR and 3-HBA ↑ and V, Y, P, K, L ↓	[56]
MS	IMS and MS/MS	Spatially resolved information	Tissue	7 PTC vs. 7 CTR from the same participants	PC (16:0/18:1) and (16:0/18:2) and SM (d18:0/16:1) ↑	[71]
	IMS and MALDI-FTIR MS		Tissue and serum	Tissue: 16 MN vs. 5 BN and 15 CTR Serum: 124 MN vs. 43 BN and 122 CTR	MN vs. BN: PA (36:2), (36:3), (38:3) ↑ PA (38:4), (38:5), (40:5) ↓	[72]
	DESI-MS	Tissue	Tissue	8 PTC vs. 18 CTR lymph nodes from the same participant	Q in adjacent lymph node, GSH, CDL, PI, PS and CER↑	[73]
	AFADESI-IMS		Tissue	12 PTC vs. 12 CTR from the same participant	F, L, Y ↑; CRE ↓	[74]
	GC-MS	Tissue extracts	Tissue extracts	16 PTC vs. 16 CTR from the same participant	MLO, IN, CHL and ARA altered; GLU, FRU, GAL, MAN, 2-keto-D-GLA and RHA ↓	[76]
	GC-TOF-MS and UHPLC-qTOF-MS		Tissue extracts	57 PTC vs. CTR from the same participant; 48 BN vs. CTR from the same participant	LACA, TCA cycle intermediates, Aa, one-carbon metabolism ↑, disrupted W metabolism in PTC and BN. TAU and HTAU and ECDA ↑ in only PTC	[109]
	GC-TOF-MS and UHPLC-QqQ-MS		Tissue extracts	Untargeted: 15 PTC vs. 15 CTR from the same participants; Targeted: 10 PTC vs. 10 CTR from the same participants	GOL, MLB and MEL ↓	[77]
	GC-MS		Formalin-fixed tissue	7 FTC, 4 PTC, 4 PTC-FV, 6 MTC, 6 ATC, 3 FA, 5 CTR	Cancerous thyroid vs. normal tissue: LACA ↑; several FA and their esters ↓. MN vs. BN: myo-IST Ph, SCA and certain FA and their esters ↑; PTC vs. follicular thyroid lesions: CTA ↑; GLA ↓	[78]
	MALDI-Q-Ion Mobility-TOF-MS		Formalin-fixed tissue sections	3 PTC vs. 3 BN from the same participant	PC (32:0), (32:1), (34:1) and (36:3), SM (34:1) and (36:1) and PA (36:2) and (36:3) ↑	[75]
	GC-MS	Systemic profiling	Exhaled breath	39 PTC vs. 25 BN and 32 CTR	PTC vs. BN: 1, 1, 3-triMe-3-(2-Me-2-propenyl) CPT, trans-2-dodecen-1-ol ↑; (3-Me-oxiran-2-yl)-methanol ↓; PTC vs. CTR: PHN, ETG mono vinyl ester, CPR, 1-bromo-1-(3-Me-1-pentenylidene)-2,2,3,3-tetraMe CPR ↑; CHX, 4-HBA, 2,2-dimethyldecane, ETH ↓	[93]
	GC-MS		Plasma	19 PTC vs. 16 BN and 20 CTR	PTC vs. BN: SUC ↑; PTC vs. CTR: E, α-KTG, AD-5 monoPh, 3-HBA, CPA, URA ↑; CYS, C↓	[83]
	nUHPLC-ESI-MS/MS		Plasma	10 TC vs. 74 other cancers and 20 CTR	TC vs. other cancers and CTR: Lyso PI (18:0) and (18:1)	[91]
	LC-LTQ Orbitrap MS		Serum	30 PTC vs. 80 BN and 30 CTR	FA, AC, SPG (SPG, SPG-1-Ph), OLM and 3-HBA ↑	[52]
	GC-TOF-MS		Serum	37 PTC-DM vs. 40 PTC-AB	N, GABA, AOA, 4- DOP ↑; PGA ↓	[79]
	LC-DIA-MS		Serum	30 PTC vs. 27 CTR	392 significantly changed metabolites	[68]
	UPLC-QTOF-MS		Fecal matter	15 TC vs. 15 CTR	3,7,11,15-tetraMe-6,10,14-hexadecatrien-1-ol, TGL (16:0/16:1(9Z)/18:2(9Z, 12Z)), 10-propyl-5,9-tridecadien-1-ol ↑; DHEAS, EPKSI ↓	[92]
	HUPLC/UHPLC-MS		Serum and urine	124 PTC vs. 76 BN and 116 CTR	PTC vs. BN and CTR: Serum β-HBA, DHA, 1-MeAD ↑, pregnanediol-3-GLC, urinary NIC mononucleotide and XNTO ↓	[89]
	UPLC-Q/TOF-MS	Tissue and systemic profiling	Tissue, serum and plasma	141 PTC vs. 93 BN and 100 CTR plus validation sets in 6 independent centers	PTC vs. CTR: Serum: 17 significantly changed metabolites; Plasma: 42 significantly changed metabolites, such as PB, L-E ↑; myo-IST, alpha-N-phenylacetyl-L-Q, lyso PC (18:0) and (18:1) ↓ PTC vs. BN: No significant differences in serum/plasma; Tissue: 16 significantly changed metabolites	[90]
	GC-MS	Culture cells	Thyrospheres with cancer stem-like cells	Cancer thyrospheres vs. cancer parental adherent cells and to non-cancer thyrospheres	SCA, MLI, D, E ↑; GLU, PYR, FRU ↓	[80]
NMR and MS	^1^H-NMR and GC−FID/MS	Tissue extracts	Tissue extracts	53 thyroid lesions vs. 46 CTR from the same participant	Ch, PC, GPC, PEA, LAC, GSH, TAU, myo- and scyllo-IST, IN, FUM, URD and Aa ↑; Lip ↓	[59]
Other	FT-Raman	Tissue	Tissue	6 MN vs. 10 BN	T3 and T4 hormones ↑	[43]
	Hyperspectral Raman microscopy	Tissue extracts	Single cells	5 PTC vs. 5 BN	Lip; Nuc ↑; F, W, Prot, ↓	[42]
	Capillary electrophoresis	Systemic profiling	Urine	12 TC vs. 12 CTR	IN, N^2^-MG, N^2^,N^2^-DMG, 1-MG ↑	[87]
	Amino acid analyser		Plasma	33 TC vs. 137 CTR	M, L, Y and K ↑	[70]

Study design abbreviations: AB, ablation; ATC, anaplastic thyroid carcinoma; BN, benign; CTR, healthy controls; DM, distant metastasis; FA, follicular adenomas; FTC, follicular thyroid carcinoma; LNM, lymph node metastasis; MN, malignant; MTC, medullary thyroid carcinoma; NA, not applicable; PTC, papillary thyroid carcinoma; PTC-FV, papillary thyroid carcinoma follicular variant; PTMC, papillary thyroid microcarcinoma; TC, thyroid carcinoma. Altered metabolites abbreviations: ACT, Acetone; ACM, Acetamide; ALC, Acetylcarnitine; AC, Acylcarnitine; AD, Adenosine; A, Alanine; Aa, Aminoacids; AOA, Aminooxyacetic acid; ARA, Arachidonic acid; N, Asparagine; D, Aspartate; CPA, Capric acid; CDL, Cardiolipin; CER, Ceramide; CHL, Cholesterol; Ch, Choline; CIT, Citrate; CTA, Citric acid; CRE, Creatine; CRN, Creatinine; CHX, Cyclohexanone; CPT, Cyclopentane; CPR, Cyclopropane; C, Cysteine; CYS, Cystine; DHEAS, Dehydroepiandrosterone sulfate; DOP, Deoxypyridoxine; DMG, Dimethylguanosine; DHA, Docosahexaenoic acid; DLC, Dolichol; ECDA, Eicosadienoic acid; EPKSI, Epimedokoreanoside I; ETA, Ethanolamine; ETG, Ethyleneglycol; ETH, Ethylhexanol; FA, Fatty acids; FRU, Fructose; FUM, Fumarate; GAL, Galactose; GOL, Galatinol; GABA, Gamma-aminobutyric acid; GLA, Gluconic acid; GLU, Glucose; GLC, Glucuronide; E, Glutamic acid; Q, Glutamine; GTA, Glutaric acid; GSH, Glutathione; GPC, Glycerophosphocholine; G, Glycine; HIP, Hippurate; HoC, Homocysteine; HoS, Homoserine; HBA, Hydroxybutyric acid; HTAU, Hypotaurine; HPX, Hypoxanthine; IN, Inosine; IST, Inositol; I, Isoleucine; KTG, Ketoglutarate; KYN, Kynurenine; LAC, Lactate; LACA, Lactic acid; L, Leucine; Lip, Lipids; LDL, Low density lipoprotein; K, Lysine; MLI, Malic acid; MLO, Malonic acid; MAN, Mannose; MEL, Melatonin; MLB, Melibiose, M, Methionine; Me, Methyl; MG, Methylguanosine; NIC, Nicotinic acid; Nuc, Nucleic acids; OLM, Oleamide; PHN, Phenol; F, Phenylalanine; Ph, Phosphate; PA, Phosphatidic acids; PC, Phosphatidylcholine; PE, Phosphatidylethanolamine; PI, Phosphatidylinositol; PS, Phosphatidylserine; PEA, Phosphoethanolamine; P, Proline; PB, Proline betaine; PPN, Propionate; Prot, Proteins PGA, Pyroglutamic acid; PYR, Pyruvate; RHA, Rhamnose; S, Serine; SM, Sphingomyelin; SPG, Sphingosine; SCA, Succinic acid; SUC, Sucrose; TAU, Taurine; TGL, Triglyceride; W, Tryptophan; Y, Tyrosine; URA, Uracil; URD, Uridine; V, Valine; VLDL, Very low density lipoprotein; XNT, Xanthine; XNTO, Xanthosine.

## 3. Conclusions

Thyroid cancer incidence has dramatically increased worldwide in recent years [110]. The need for reliable biomarkers that can be used for fast and accurate diagnosis of the disease is critical since the initial cytological diagnostic evaluation via FNAB often provides indeterminate results and distinguishing between different types of thyroid nodules is uncertain. Several “omics” approaches have been applied to the study of thyroid cancer, but it has only been in the past few years that metabolomics has come to the forefront. A single analytical technique is not capable of thoroughly profiling the entire metabolome, but NMR spectroscopy and mass spectrometry-based approaches provide complementary information on a wide range of metabolites. At the present time, the majority of metabolomic studies have focused on core biopsies taken after inconclusive FNABs; hence, at first glance, it may be argued that these studies have not provided a complementary technique to FNAB so that a surgical procedure could be prevented. However, the fact that distinctive metabolite signatures have been found for benign versus malignant thyroid lesions coupled with identifiable metabolite profiles in body fluids suggests that metabolite profiling at the initial stage could contribute to a more confident diagnosis. Before these techniques are applied in a clinical context, it will be imperative to validate them. The application of standardised procedures for metabolomic studies, as well as improvements in the identification of metabolites by bioinformatic tools, will also be important. In metabolomics, more than in other “omics”, the collaboration and integration of clinicians, biologists, chemists, statisticians and bioinformaticians to obtain the most out of the ongoing research is pivotal. Finally, the integration of other “omics” techniques along with metabolomics is still lacking, in particular the use of miRNAs could ameliorate the diagnostic power of solely metabolite panels. “Omics” collaborations will be essential in order to fully understand this disease and to discover new therapeutic targets or diagnostic biomarkers.

In a clinical setting, we will more likely see mass spectrometers than an NMR spectrometer. Both LC-MS and NMR spectrometers can have a large initial price at installation, but GC-MS instruments are more affordable, although in terms of cost per sample, NMR can provide analysis at lower costs than the other two MS techniques. On the other hand, NMR spectroscopy is known for its reproducibility, capability for absolute quantification and comparability of spectra inter-laboratories, but these hurdles can be overcome in MS by the use of internal standards to monitor reproducibility, the use of analytical standards for absolute quantification and standard operating procedures for inter-laboratory comparison. Moreover, mass spectrometers require less specialised space for the equipment and are capable of measuring low-concentration metabolites. The development of new analytical techniques (e.g., ion mobility MS and mass imaging) and the improvement of data analysis tools are increasing our knowledge regarding the metabolic changes and their role at both systemic and cellular levels in different pathological conditions. This information can also be acquired with a smaller amount of tissue or blood nowadays. Presently, a better understanding of the metabolome may support the diagnosis, better define the staging and the prognosis of cancer, too. By monitoring the metabolic changes, metabolomics may also predict the response to therapy and possible side effects of it. In the next years, we can expect that the development of metabolomics will give a big contribution towards a more precise and personalised approach to thyroid cancer in a systemic context. Not only will thyroid biopsies be used, but relevant information will be achieved by the integration of omics data from blood samples, considering it as a liquid biopsy of the tumour.

## Figures and Tables

**Figure 1 ijms-21-05272-f001:**
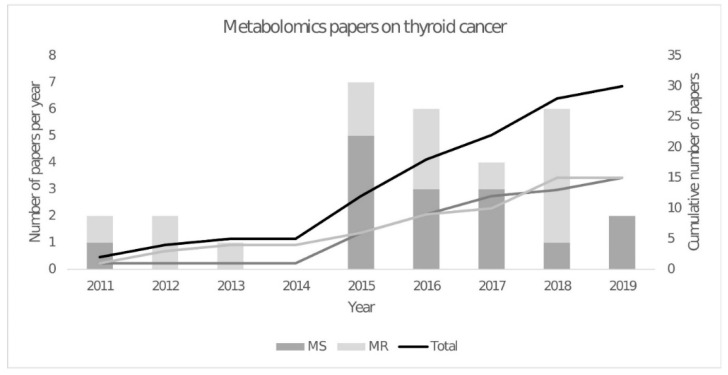
Number of papers of untargeted metabolomics studies in thyroid cancer using magnetic resonance spectroscopy and high-resolution liquid and solid state NMR spectroscopy (MR) and mass spectrometry (MS). Papers found in the PubMed and Web of Science on April 16th 2020. Criteria—Pubmed: (((thyroid neoplasms[MeSH Terms]) OR (metabolomic*[MeSH Terms])) AND (metabolom*[MeSH Terms])) AND (thyroid[Title/Abstract]) Filters: Humans, English and (thyroid[Title/Abstract]) AND ((cancer*[Title/Abstract]) OR (carcinom*[Title/Abstract]) OR (malignant[Title/Abstract])) AND ((metabolom*[Title/Abstract]) OR (metabolit*[Title/Abstract])) Filters: Humans, English. Web of Science: ((TI = (thyroid AND (cancer OR carcinom* OR neoplasm* OR malignant*) AND (metabolomic* OR metabonom* OR metabolit*)))) AND English AND Article. Note: Reviews, other non-related papers, response to treatment or other omics studies that were not untargeted metabolomics were excluded.

**Figure 2 ijms-21-05272-f002:**
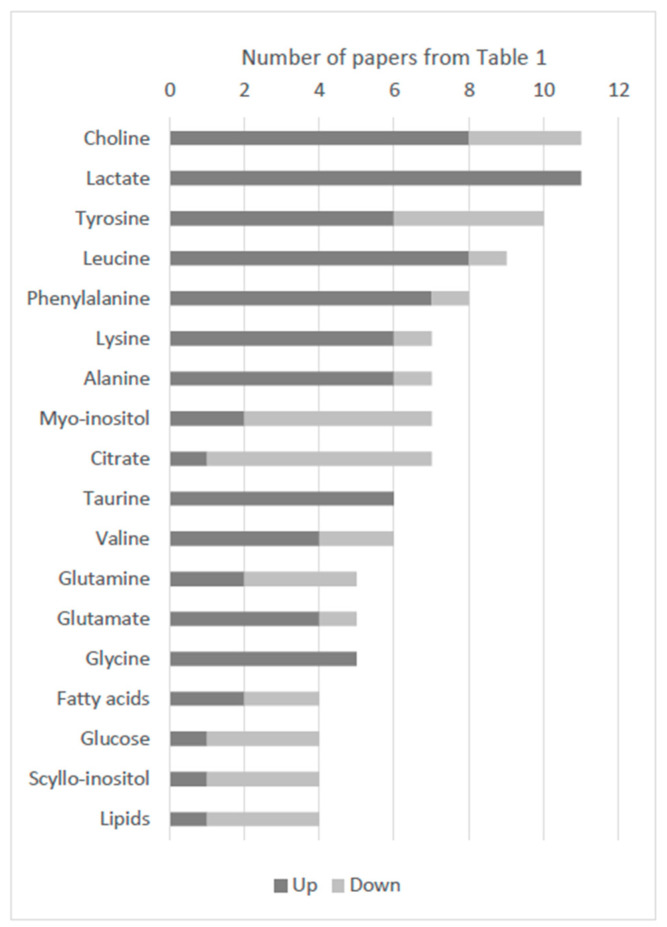
Metabolites featured in thyroid cancer versus healthy or benign controls. Metabolites referenced more than three times in the metabolomic studies of thyroid cancer showcased in Table 1, altered or with discriminative value. Dark grey, upregulated; light grey, downregulated.

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
