# Peer review of "The Potential of Metabolomics in the Diagnosis of Thyroid Cancer"

_ijms, 2020, doi:10.3390/ijms21155272_

Round 1

Reviewer 1 Report

The paper “The potential of metabolomics in the diagnosis of thyroid cancer” is a narrative review about the knowledge of metabolomics of thyroid cancer. This aspect is of great interest since in the last 20 years both proteins expression and genetic analysis failed to find a definite biomarker able to discriminate benign from malignant thyroid nodules. The review covers different methods of analysis and shows heterogeneous metabolic profiles of the disease.

The paper is well written and the results of the analyzed studies are clearly reported.

Anyway, spectroscopic techniques such as Raman spectroscopy have been not considered in this paper. Raman spectroscopic has emerged as a tool for thyroid tissue analysis from at least ten years and it revealed biochemical and molecular changes with diagnostic potential (for example: C. S. B. Teixeira, et al. Thyroid tissue analysis through Raman spectroscopy. Analyst, 2009, 134, 2361–2370; M. A. S. De Oliviera, et al. Hyperspectral Raman microscopy can accurately differentiate single cells of different human thyroid nodules.  Biomedical Optics Express Sep 2019 Vol. 10, No. 9). A sentence about these results should be added in the paper for a complete overview.

Author Response

Response to Reviewer 1 Comments

The paper “The potential of metabolomics in the diagnosis of thyroid cancer” is a narrative review about the knowledge of metabolomics of thyroid cancer. This aspect is of great interest since in the last 20 years both proteins expression and genetic analysis failed to find a definite biomarker able to discriminate benign from malignant thyroid nodules. The review covers different methods of analysis and shows heterogeneous metabolic profiles of the disease.

The paper is well written and the results of the analyzed studies are clearly reported.

Response: We thank the reviewer for these positive remarks

Point 1: Anyway, spectroscopic techniques such as Raman spectroscopy have been not considered in this paper. Raman spectroscopic has emerged as a tool for thyroid tissue analysis from at least ten years and it revealed biochemical and molecular changes with diagnostic potential (for example: C. S. B. Teixeira, et al. Thyroid tissue analysis through Raman spectroscopy. Analyst, 2009, 134, 2361–2370; M. A. S. De Oliviera, et al. Hyperspectral Raman microscopy can accurately differentiate single cells of different human thyroid nodules.  Biomedical Optics Express Sep 2019 Vol. 10, No. 9). A sentence about these results should be added in the paper for a complete overview.

Response 1: We added a sentence about these results, as suggested by the reviewer, as well as a note later in the text (Page 4, 2. Metabolomics in Thyroid Cancer, lines 153-157; Page 12, 2.4. Most referred metabolites, line 448). In addition, these references were included in Table 1, as well as in Figure 2.

Reviewer 2 Report

The review by Coelho et al gives a comprehensive picture on the current state of knowledge regarding the

metabolomics as tool in the study of thyroid cancer.

In particular, the review discusses NMR and GC-MS applications with respect to the identification of potential diagnostic  biomarkers in thyroid cancer.

However, with respect to the NMR and GC-MS applications, other very recent references could be included. In detail, the paper by Metere et al., “Metabolomic Reprogramming Detected by 1H-NMR Spectroscopy in Human Thyroid Cancer Tissues” Biology, 9:6(2020), p. 112 and the paper by Abooshahab et al, “Plasma Metabolic Profiling of Human Thyroid Nodules by Gas Chromatography-Mass Spectrometry (GC-MS)-Based Untargeted Metabolomics”, Front Cell Dev Biol. 2020; 8: 385, could be included.

The review discusses changes reported for metabolites  levels (particularly choline, tirosine and lactate) in thyroid cancer during tumor transformation and progression.

However in my opinion the authors should better interpret the reported changes with regard to underlying molecular mechanisms linked to the biosynthetic and catabolic pathways.

Additionally, the authors could mention other examples of NMR metabolomics as a source of diagnostic surrogates  of thyroid carcinoma.

For example, , Li  et al (Metabolic Changes Associated With Papillary Thyroid Carcinoma: A Nuclear Magnetic Resonance-Based Metabolomics Study. Int J Mol Med . 2018 May;41(5):3006-3014) and Gupta et al, (Magnetic resonance spectroscopy as a diagnostic modality for carcinoma thyroid, Eur J Radiol. 2007;64(3):414–418), demonstrate its power in the diagnosis of thyroid cancer.

I suggest to integrate the refereces in figure 1-2 and table as well.

Author Response

Response to Reviewer 2 Comments

The review by Coelho et al gives a comprehensive picture on the current state of knowledge regarding the metabolomics as tool in the study of thyroid cancer.

In particular, the review discusses NMR and GC-MS applications with respect to the identification of potential diagnostic biomarkers in thyroid cancer.

Point 1: However, with respect to the NMR and GC-MS applications, other very recent references could be included. In detail, the paper by Metere et al., “Metabolomic Reprogramming Detected by 1H-NMR Spectroscopy in Human Thyroid Cancer Tissues” Biology, 9:6(2020), p. 112 and the paper by Abooshahab et al, “Plasma Metabolic Profiling of Human Thyroid Nodules by Gas Chromatography-Mass Spectrometry (GC-MS)-Based Untargeted Metabolomics”, Front Cell Dev Biol. 2020; 8: 385, could be included.

Response 1: These references were added to the document (Page 5, 2.1. The early years – NMR spectroscopy, lines 204-206; Page 8, 2.3. Peripheral fluids, lines 353-356). In addition, these references were included in Table 1, as well as in Figure 2.

Point 2: The review discusses changes reported for metabolites levels (particularly choline, tyrosine and lactate) in thyroid cancer during tumor transformation and progression.

However in my opinion the authors should better interpret the reported changes with regard to underlying molecular mechanisms linked to the biosynthetic and catabolic pathways.

Response 2: We have attempted to address the reviewers concerns by adding a few sentences regarding common pathways of some of the most referenced metabolites, but not specifically the first three (Page 11, 2.4. Most referred metabolites, lines 423-429). We must stress that there is severe lack of information on whether the biochemical change is directly related with the disease or is a response of the organism to the disease. Therefore, attempting to underly the molecular mechanisms from the available information might be misleading. These metabolites should be studied in translational models to address if they are associated with the molecular mechanisms of the disease or if they are just a response of the organisms to the presence of the disease.

Point 3: Additionally, the authors could mention other examples of NMR metabolomics as a source of diagnostic surrogates of thyroid carcinoma.

For example, , Li  et al (Metabolic Changes Associated With Papillary Thyroid Carcinoma: A Nuclear Magnetic Resonance-Based Metabolomics Study. Int J Mol Med . 2018 May;41(5):3006-3014) and Gupta et al, (Magnetic resonance spectroscopy as a diagnostic modality for carcinoma thyroid, Eur J Radiol. 2007;64(3):414–418), demonstrate its power in the diagnosis of thyroid cancer.

Response 3: The reference by Li et al. was already mentioned in the manuscript and included in Table 1 (Page 5, 2.1. The early years – NMR spectroscopy, lines 211-212). In the case of the paper by Gupta et al. we already had a similar paper by the same authors (Gupta, N.; Goswami, B.; Chowdhury, V.; RaviShankar, L.; Kakar, A. Evaluation of the role of magnetic resonance spectroscopy in the diagnosis of follicular malignancies of thyroid. Arch Surg 2011, 146, 179-182, doi:10.1001/archsurg.2010.345) with similar results. In this case we have added the reference suggested by the reviewer to the reference that was already present in the manuscript, without changing the text (Page 4, 2.1. The early years – NMR spectroscopy, lines 168 and 174). In addition, this new reference was included in Table 1, as well as in Figure 2.

Point 4: I suggest to integrate the refereces in figure 1-2 and table as well.

Response 4: We have integrated all the references suggested by both reviewers into Table 1 and Figure 2, however we did not updated the information on Figure 1 because we had a different approach in the making of this graph. While Figure 2 information was based on the content extracted from Table 1, Figure 1 was based on papers found during a specific bibliographic search as described in the Figure 1 legend, therefore little changes would have been made to this figure if it was repeated, since the last search date was on April 16th 2020. In addition, we do not want to include the newly published papers from the year 2020 to not be misleading, since the year is not over yet. Moreover, the papers found in this search do not necessarily coincide with the papers present in Table 1 since other papers might be found by looking up other cited references and not necessarily by using search engines. However, we felt that to represent these numbers it would be better to use well-defined criteria to overcome the bias present in the selection of papers made by the own authors subjectivity. We do not aim to be a systematic review, but rather a literature review, and we do mention that the results obtained in this Figure 2 are a reflection of the studies we presented (Page 12, lines 429).